# Comparison of Knee and Hip Kinematics during Landing and Cutting between Elite Male Football and Futsal Players

**DOI:** 10.3390/healthcare9050606

**Published:** 2021-05-18

**Authors:** Abdolhamid Daneshjoo, Hadi Nobari, Aref Kalantari, Mohammadtaghi Amiri-Khorasani, Hamed Abbasi, Miguel Rodal, Jorge Pérez-Gómez, Luca Paolo Ardigò

**Affiliations:** 1Department of Sports Injuries and Corrective Exercises, Faculty of Sports Sciences, Shahid Bahonar University of Kerman, Kerman 7616913439, Iran; daneshjoo.hamid@uk.ac.ir (A.D.); aref07kalantari@gmail.com (A.K.); 2HEME Research Group, Faculty of Sport Sciences, University of Extremadura, 10003 Cáceres, Spain; Jorgepg100@gmail.com; 3Sports Scientist, Sepahan Football Club, Isfahan 81887-78473, Iran; 4Department of Physical Education and Sports, University of Granada, 18010 Granada, Spain; 5Department of Exercise Physiology, Faculty of Sport Sciences, University of Isfahan, Isfahan 81746-7344, Iran; 6Department of Biomechanics, Faculty of Sports Sciences, Shahid Bahonar University of Kerman, Kerman 7616913439, Iran; amirikhorasani@uk.ac.ir; 7Department of Sport Injuries and Corrective Exercises, Sports Sciences Research Institute, Tehran 1587958711, Iran; h.abbasi@ssrc.ac.ir; 8BioErgon Research Group, University of Extremadura, 10003 Cáceres, Spain; mrodal@unex.es; 9Department of Neurosciences, Biomedicine and Movement Sciences, School of Exercise and Sport Science, University of Verona, 37131 Verona, Italy

**Keywords:** hip flexion, knee flexion, knee valgus angle, risk of injury, prevention, cutting maneuver

## Abstract

To design an accurate sport injury prevention program, alterations in the knee and hip kinematic variables involved in injury mechanisms should be known. The main purpose of the current study was to compare knee and hip kinematic variables during landing and cutting among male football and futsal players, and to discuss them within an injury description frame. Twenty football (20.5 ± 2.1 years., 74.5 ± 6.9 kg and 1.79 ± 0.07 m) and twenty futsal players (20.3 ± 2.0 years., 73.5 ± 7.1 kg and 1.78 ± 0.07 m), with at least three years’ experience of playing in the Kerman Province League, participated in this study. Hip flexion, knee flexion and knee valgus angle during two main movements with risk of injury, such as landing and cutting, were measured using a motion capture system with passive markers at 120-Hz sampling frequency. Landing and cutting maneuvers were administered in as natural way as possible. Results showed significant differences in landing and cutting maneuvers between groups in hip flexion, knee flexion and knee valgus angle. Results indicated that footballers have less extension of hip and knee joints than futsal players in landing maneuvers, which may be due to the higher requirement of jumping−landing maneuvers when playing football. In cutting maneuvers, footballers showed less hip and knee flexion than futsal players, whereas the knee valgus angle in cutting maneuvers was lower in futsal players. More information on the injury mechanisms of landing and cutting in football and futsal are needed to improve the design of injury prevention programs.

## 1. Introduction

Football in the 21st century is one of the most popular sports in the world, because it is played by 300 million people from 203 different nations at amateur or recreational level [1]. Futsal is the five-a-side version of football and is played on a smaller pitch (width 15–25 m, length 38–42 m). It is played by over a million registered players worldwide, and it is a growing sport in many countries. Futsal is a multiple-sprint sport with more high intensity phases than football as witnessed by the higher relative number of injuries in futsal occurring during noncontact activities compared with football (37% vs. 27% [2]). In comparison with football, where there are many publications about kinematic variables as injury risk predictors, a few scientific studies have been published about it on futsal players [3,4].

Unfortunately, football and futsal are associated with a high risk of injuries in the lower extremities, which results in significant costs for the public health system and may even cause long-term disability for the injured player. The costs associated with surgically reconstructed anterior cruciate ligament (ACL) injuries range from $5000 to $17,000 per player. However, the estimated long-term societal costs may be as high as $38,000 per player [5]. Commonly, the players are allowed to return to sporting activities, including cutting and jumping, six months after ACL injuries. There is a high risk of long-term osteoarthritis in all treated populations following ACL injury [6]. Knowing the reasons and mechanisms of the injuries is a critical step to design injury prevention programs. In order to design an accurate prevention program, the alterations in kinematic variables should be known for reference [7]. Generally, most of the injuries in football and futsal occur in the lower extremities, and the most common noncontact injuries are lower limb injuries after jumping and landing [8]. Hawkins et al. [9] found that the percentage of noncontact mechanisms of injury was higher than contact mechanisms injuries, and one of the most common noncontact mechanisms in football was jumping and landing. Similarly, Pollard et al. [10] reported that 70% of the ACL injuries occurred due to noncontact mechanisms, and most of these injuries occurred during dynamic maneuvers such as landing, shear movements, acceleration and direction changes. Park et al. [11] stated that unexpected and unplanned landing and unloading maneuvers could increase the risk of injury.

In addition to contact in football, this sport includes noncontinuous sprint during the match. The high-speed maneuver and tough intensity in futsal compared to football are some of the main reasons that can explain the higher injury risk factor in futsal. However according to the literature, the amount of lower limb injuries in football are 2.5 times more than futsal [2].

Minetti et al. [12] investigated the use of muscles as power dissipators both by model and experimentally. The modelled predictions of the drop landing maneuver, similar to the footballers’ landing, for an initial condition span were calculated, considering the mechanical features of knee extensor muscles, the lower limb arrangement and hypothesizing a maximal neural activation. The resulting dynamics was shown in the form of a phase−plane graph (viz., vertical displacement vs. speed) to functionally feature the damping performance. The modelled estimate of a “safe” (in terms of dynamics, i.e., with the subject’s maximum available braking torque ≤500 N·m; see Figure 9 from Minetti et al. [12]) landing was shown to happen when falling from a maximal height of 1.6–2.0 m in sedentary subjects. Differently, athletes resembling trained footballers can safely landing from a maximal height of 2.6–3.0 m. Such results were calculated by considering within the model of the literature-estimated stiffness of the lower limbs’ all elastic elements working below the squat position height. Experiments consisted of landings from heights of 0.4, 0.7 and 1.1 m (thus similar to normal football training and match heights) showing dynamics similar to the model. Namely, peak vertical power landing from 0.7 m results were similar between sedentary subjects and female athletes, whereas male athletes’ values results were 40% greater. As a consequence, male athletes (i.e., with physical capacities similar to trained footballers) stopped the landing downward movement 20% faster than sedentary subjects, reducing the risk of injury to lower limb structures. Overall, regarding landing, “safety” vertical displacement vs. speed zones are already known in athletes (Figure 10 in Minetti et al. [12]).

In several studies, the kinematic variables featuring the lower extremities have been assessed within the search of risk factors. For example, Blackburn and Padua [13] stated that extended posture of knee and hip and greater knee valgus angle during landing were associated with ACL injury. Griffin et al. [14] also related knee valgus angle and the extended posture of knee, hip and vertebral column to risk factors for ACL injury. Tsai et al. [15] provided justification for injury prevention programs by concluding that greater hip and knee flexion during landing can reduce tibiofemoral joint loading and consequently decrease ACL injuries.

As mentioned above, there are some differences between football and futsal, but a futsal player generally follows injury prevention programs that are designed based on football. There are some studies that examined kinematic variables featuring different team sports [16]. Meanwhile, to our knowledge, there is no research that has compared the kinematic variables of the lower limbs during two common injury mechanisms, landing and cutting maneuvers, among football and futsal players. On the other hand, the safest and most appropriate method of performing landing and cutting maneuvers has not yet been identified. Such findings could possibly be a way to prevent injury over those maneuvers. For these reasons and due to the fact that football and futsal show different game paces (also given their different pitch sizes), it was hypothesized that football and futsal would likely show differences in kinematic variables featuring some specific injury-risk sport fundamentals. Therefore, the present study aims at comparing the kinematic variables of knee and hip during landing and cutting among football and futsal players, and to discuss them within an injury description frame.

## 2. Materials and Methods

All subjects were informed verbally about the procedures of the study and their written consent was obtained. The study was approved by the Ethics Committee of Shahid Bahonar University of Kerman (protocol code 021.95.658 and date of approval 18 January 2017). In this study, the Helsinki recommendations for human studies were followed over all stages.

Twenty male footballers (mean ± standard deviation [SD], age 20.5 ± 2.1 years., mass 74.5 ± 6.9 kg and height 1.79 ± 0.07 m) and twenty male futsal players (age 20.3 ± 2.0 years., mass 73.5 ± 7.1 kg and height 1.78 ± 0.07 m) participated in this study. No differences were found between football and futsal player in any demographic variables (*p* > 0.05). All subjects—with at least 3 years’ experience in the Kerman Province League—were participating in football clubs with similar training schedules that included practice or competition at least 3 times per week (60–90 min per session). Subjects reported no lower extremity injury within the 6 months before data collection, and they were free of pain at the time of the study. Subjects with previous knee surgery, ligamentous instability or any medical or neurologic condition that would impair their ability to perform the experimental tasks were excluded. Before participation, all procedures were explained to each subject. On a separate occasion before the test session, all the players attended a workshop to learn the proper ways to carry out the tests, i.e., up to feeling comfortable with undergoing them. All tests were conducted at the Biomechanics Research Laboratory at the Shahid Bahonar University of Kerman, on a normal indoor surface with temperature 20–24 degrees Celsius. All tests were carried out between 8 a.m. and 11 a.m. Each subject underwent one single test session.

Before data collection, height and mass were measured. These measurements were made by a person with more than 6 years of experience in anthropometric measurements [17,18,19]. The considerations were performed based on previous research [20,21,22,23,24]. Afterwards 18 reflective markers (19-mm spheres), according to the Helen Hayes model, were bilaterally attached to the following bony landmarks: distal first toe, fifth metatarsal head, lateral malleoli, shank, tibial tuberosity, lateral epicondyle of femur, thigh, greater trochanter and anterior superior iliac spine. Markers were attached with adhesive tape directly to the subject’s skin [25]. All the markers were attached each time by the same researcher. Regarding motion analysis, kinematic data (i.e., markers’ positions over time) were collected using a Raptor-H six-camera (Motion Analysis System, Cortex, Rohnert Park, CA, USA), and a three-dimensional motion analysis system (Raptor-H Digital Real Time System) at a sampling frequency of 120 Hz. Before data collection, subject performed 10 min dynamic stretching with special attention to the lower body, including calf muscles, quadriceps, hamstrings, hips and shoulders. The same researcher managed all recording trials.

All subjects were asked to perform two tasks: the landing and the side cutting. Subjects were asked to perform three test trials for each task along a 10-m distance. Between each trial and the following one, subjects were allowed to rest at least 5 min. The order of testing was randomized. All subjects wore indoor sport shoes.

Regarding the landing maneuver, the task was designed similarly to a real situation in the pitch. Preliminarily, 50% of the maximum jump height was measured using a digital Sergeant machine (JS-D80, Yagami, Tokyo, Japan) with high reliability (intraclass correlation coefficient = 0.85 [26]). Maximum jump height (51–70 cm) was recorded after three trials, and best score was used for further analysis. After marker-attaching and preparing subjects, the examiner hung a ball at a height appropriate—i.e., similar to a real pitch situation—for the participants (ball jump height = height of participant + 50% of maximum jump height). Then, the subjects were asked to perform jumping, heading and landing as performed during practice and competition. For this purpose, the subjects were placed at a distance of 5 m from the ball (i.e., a reasonable run-up distance just before a heading after, e.g., a corner). After running and going to the intended place, the subjects jumped, performed heading and then landed. Each subject performed three trials, and the best performance with maximum jump height was used for the analysis. Average of variable values of both lower limbs were used for the analysis.

Regarding the cutting maneuver (Figure 1), subjects were instructed to run at medium-to-fast pace throughout 7 m until touching with their right foot the change of direction area, and then to change the direction to the left (for left footed it was the opposite). Range between the angles of 35° and 55° with respect to the initial path was found and marked using a goniometer and two cones on the floor, and subjects were instructed to perform the change of direction at the angle of 45° [27]. Each subject performed three trials with one minute of rest between attempts and the best performance was used for the analysis. Maximum knee and hip angle of dominant lower limb during this cutting maneuver at 45° were used for the analysis.

Regarding the statistical analysis, the data were analyzed using the Statistical Package for Social Sciences (IBM SPSS, version 21, Armonk, NY, USA) and were presented as mean and SD. The data were also screened to ensure that assumptions of normality and independence were met for statistical analysis. Namely, the Shapiro−Wilk and the runs tests were used to assess the normality of data distribution and the hypothesis of independence, respectively (*p* > 0.05). Independent *t*-test was used to compare groups. Significance was accepted at the 95% confidence level for all statistical parameters (*p* < 0.05).

## 3. Results

The means and SDs of the kinematic variable values from the landing maneuver are presented in Table 1. The results showed significant differences between groups during the landing maneuver in maximum hip flexion, maximum knee flexion and maximum knee valgus angle (*p* < 0.05).

The means and SDs of kinematic variables from cutting maneuver are displayed in Table 2. The results indicated significant differences between groups during cutting maneuver in maximum hip flexion, maximum knee flexion and maximum knee valgus angle (*p* < 0.05).

## 4. Discussion

The purpose of the present study was to compare lower extremities kinematic variables during common potential injury mechanisms such as landing and cutting among male football and futsal players. The results showed that the maximum values of hip flexion, knee flexion and valgus angle were different between football and futsal players. Footballers had greater hip and knee flexion, but less knee valgus angle during the landing maneuver—and therefore a safer landing—than futsal players. Such a difference between football and futsal is credited to the games’ natures, which imply continued bouts of intensive physical activity, high-speed execution, smaller pitch dimensions and a harder playing surface in futsal [28]. All these peculiarities combine to increase noncontact injuries in futsal [2]. Playing football requires more activities such as jumping vertically and heading the ball, because the pitch is larger and therefore long passes—and consequently “high balls”—are more frequent. These demands may get young professional soccer players acquainted with safe landing [2]. Moreover, the landing maneuver test, which was used in this study, was more similar to playing football. Decker et al. [29] reported that an erect position in landing increases energy absorption and this may be the reason why futsal players choose to land in a more erect posture, to maximize the energy absorption by the joints most proximal to ground contact [29]. However, futsal players do not perform as many landing maneuvers as footballers, probably due to the different match-play (i.e., with more “low balls” over a smaller pitch). Therefore, the landing maneuver may not be that relevant to a futsal player’s injury risk given that this action is not as usual as it is in football.

It is known that at an angle beyond 45° of knee flexion, the quadriceps and ACL perform the same action, but at knee flexion less than 45° the ACL action is in contrast with the quadriceps and this may increase the risk of knee injury. It was found that low knee valgus angle (around 8° or lower) decreases the risk of knee injuries [30]. It may be concluded that footballers are predisposed to get less injuries in landing, because their kinematics are closer to normal values and that reduces the risk of injury. The presence of a higher quantity of landings in playing football likely contributes to developing a safer landing technique.

The results indicated higher hip flexion and knee flexion in both groups in the cutting maneuver, but the knee valgus angle result was lower in the futsal players than the footballers in this maneuver. It is known that footballers, in the cutting maneuver, may be predisposed to get more injuries in the lower extremities, because it is known that a greater knee valgus angle increases knee injury likelihood, especially ACL injuries. Markolf et al. [31] showed that the mean ligament force was almost equal to the applied anterior tibial force near 30° of flexion, and up to 150 percent of applied tibial force at full extension. The addition of internal tibial torque to a knee loaded by anterior tibial force produces increments of force at full extension. This loading combination produces the highest ligament forces, and is the most dangerous in terms of potential injury to the ligament. In contrast, the addition of external tibial torque to a knee loaded by anterior tibial force decreases the force in the flexed positions of the knee. At close to 90° of flexion, the ACL becomes completely unloaded. The addition of a valgus moment increases the force in flexed positions [25,26,30,31]. These states of combined loading could also present an increased risk for injury. The similarity of the playing surface for futsal to the laboratory setting may result in less knee valgus angle in the futsal player than in the footballers. This also implies that future studies should take into account the different existing surfaces available to footballers (e.g., natural grass or artificial turf), because they may influence their comparison with futsal players. Moreover, both sports players’ actions should be investigated in more detail to eventually detect differences other than hip or knee kinematics. New findings could provide further indications to design optimal training sessions in order to prevent hip and knee injuries.

## 5. Conclusions

It can be concluded that, in the landing maneuver, the footballers show a lower hip and knee flexion and a lower knee valgus angle, which may predispose them to lower levels of risk of lower limb injury than futsal players. In the cutting maneuver, footballers show lower hip and knee flexion than futsal players. Yet, the knee valgus angle, in the cutting maneuver, occurred less in futsal players compared with footballers, and this may lead to less injuries due to the decreased knee valgus risk. The findings of this research can be helpful for coaches and trainers, who can design exact training for injury prevention for futsal and football, specifically.

## Figures and Tables

**Figure 1 healthcare-09-00606-f001:**
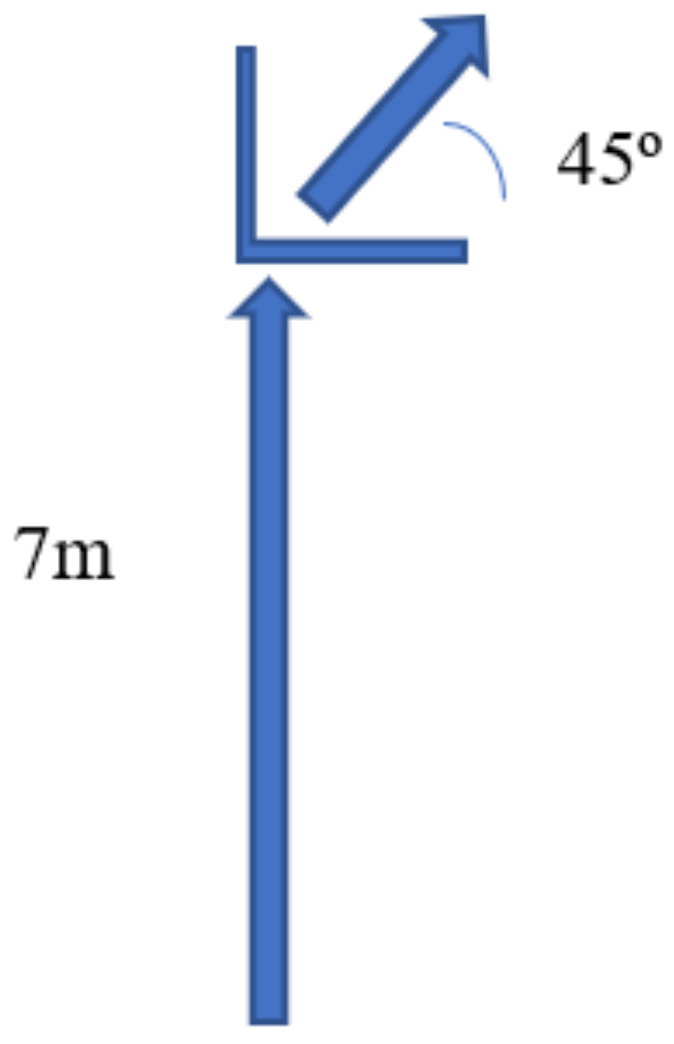
Cutting maneuver layout.

**Table 1 healthcare-09-00606-t001:** Mean ± standard deviation (SD) of kinematic variables during landing.

Kinematic Variables	Groups	Mean ± SD	Df	*t*-Test	*p*-Value
Hip flexion (max, degrees)	Football	55.63 ± 4.13	38	8.93	0.001 *
	Futsal	43.36 ± 4.53			
Knee flexion (max, degrees)	Football	70.01 ± 3.70	38	9.86	0.001 *
	Futsal	58.35 ± 3.78			
Knee valgus angle (max, degrees)	Football	6.38 ± 0.40	38	9.95	0.001 *
	Futsal	7.52 0.32			

Max = maximum. * Represents a statistically significant difference between groups (*p* < 0.05).

**Table 2 healthcare-09-00606-t002:** Mean ± standard deviation (SD) of kinematic variables during cutting.

Kinematic Variables	Groups	Mean ± SD	Df	*t*-Test	*p*-Value
Hip flexion (max, degrees)	Football	41.46 ± 3.12	38	2.64	0.012 *
	Futsal	37.18 ± 6.54			
Knee flexion (max, degrees)	Football	66.81 ± 3.25	38	34.01	0.001 *
	Futsal	35.17 ± 2.59			
Knee valgus angle (max, degrees)	Football	8.92 ± 0.50	38	11.19	0.001 *
	Futsal	7.06 ± 0.55			

Max = maximum. * Represents a statistically significant difference between groups (*p* < 0.05).

## Data Availability

The data presented in this study are available on request from the corresponding authors.

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
