# Peer review of "Comparison of Knee and Hip Kinematics during Landing and Cutting between Elite Male Football and Futsal Players"

_healthcare, 2021, doi:10.3390/healthcare9050606_

Round 1

Reviewer 1 Report

General comments: Overall, I feel that this study was well conducted. I do have some questions regarding the methodologies outlined below but I feel that these have the potential to be addressed and meet the standards for publication.

Uncertain why the words maneuver, maximum, and valgus are italicized quite frequently throughout manuscript.

Line 36 - Change to 21st century

Line 72 - I believe this is meant to be 1.6-2.0m, not the +/- symbol as that would indicate significant SD?

Line 80 - Recommend removing "ones"

Methods:

Please include a general number of repetitions that were performed for the practice trial as this could impact fatigue or potential learning effect.

How much rest was given between trials and different actions? Were they performed on the same day or different days?

Confused as to how maximum jump height was measured or what role that played in the task if they performed the jump to the height of the 50% of jump height? Consider revising to reflect the role that maximum jump height was used.

Line 141-142 - Reword sentence as the way it currently reads does not make sense. I believe rather than saying "over", the word "during" may be better suited if authors are meaning that the individuals performed movements similar to practice and competitions.

Line 146-At what intensity were they instructed to run?

Was direction of cut related to the foot preference of the individual?

It may be helpful for the readers to follow the exact task that was performed while including some type of visual schematic with the layout of the task with actions performed.

Results:

What limb was used for the maximum angles?

Line 205-What is meant by low knee valgus angle? Recommend providing a reference for what the threshold would be for valgus angle. Revise sentence as grammar is confusing. Recommend removing "that too"

Discussion:

One factor that is briefly mentioned but may be a main component is that the futsal players may not be used to the landing maneuver based on the fact they may not perform that action as often compared to the footballers. Although they may be at a greater risk for injury based on the measures you made, the action may not be relevant to the group in question and injury risk.

Were any other performance measures taken such as jump height (mentioned but not reported), sprint speed, or lower body strength? There may be differences in these values and this could be the explanation for these differences observed between groups. There was a mention of 3 session per week of practice/competition but what about any type of resistance training or current injury prevention programs?

Based on the findings, was there any follow-up with the individuals that those were highlighted as being more likely to get injured and actually getting injured during sport practice/competition?

Author Response

Does the introduction provide sufficient background and include all relevant references?

(x) Can be improved

Please, read below comments to specific points.

Is the research design appropriate?

(x) Can be improved

Please, read below comments to specific points.

Are the methods adequately described?

(x) Must be improved

Please, read below comments to specific points.

Are the results clearly presented?

(x) Must be improved

Please, read below comments to specific points.

Are the conclusions supported by the results?

(x) Must be improved

Please, read below comments to specific points.

Point 1: Uncertain why the words maneuver, maximum, and valgus are italicized quite frequently throughout manuscript.

Response 1: We thank expert reviewer for his/her suggestion. Words maneuver, maximum and valgus were changed to regular throughout manuscript.

Point 2: Line 36 - Change to 21st century.

Response 2: Suggested change was operated.

Point 3: Line 72 - I believe this is meant to be 1.6-2.0 m, not the +/- symbol as that would indicate significant SD?

Response 3: I apology for the mistake. “±” was changed to “-” throughout paragraph.

Point 4: Line 80 - Recommend removing "ones".

Response 4: “ones” was removed.

Point 5: Please include a general number of repetitions that were performed for the practice trial as this could impact fatigue or potential learning effect.

Response 5: I was not clear in my explanation. Sentence was re-worded as follows:

“On a separate occasion before the test session, all the players attended a workshop to learn the proper ways to carry out the tests, i.e., up to feeling comfortable with undergoing them.”

Point 6: How much rest was given between trials and different actions? Were they performed on the same day or different days?

Response 6: Details were added as follows:

“Each subject underwent one single test session. … Between each trial and the following one, subjects were allowed to rest at least 5 min.”

Point 7: Confused as to how maximum jump height was measured or what role that played in the task if they performed the jump to the height of the 50% of jump height? Consider revising to reflect the role that maximum jump height was used.

Response 7: Basically, our intent was to simulate an ecological setting for the landing maneuver. Details were added as follows:

“Preliminarily, 50% of maximum jump height was measured using digital Sergeant machine (JS-D80, Yagami, Japan). … After marker-attaching and preparing subjects, examiner hanged a ball at a height appropriate – i.e., similar to a real pitch situation – for participants (ball jump height= height of participant+50% of maximum jump height).”

Point 8: Line 141-142 - Reword sentence as the way it currently reads does not make sense. I believe rather than saying "over", the word "during" may be better suited if authors are meaning that the individuals performed movements similar to practice and competitions.

Response 8: “over” was changed to “during”.

Point 9: Line 146 - At what intensity were they instructed to run?

Response 9: Detail was added as follows:

“Regarding cutting maneuver (Figure 1), subjects were instructed to run at medi-um-to-fast pace throughout 7 m until touching with their right foot the change of direction area and then to change the direction to the left (for left footed it was the opposite).”

Point 10: Was direction of cut related to the foot preference of the individual?

Response 10: Yes, as explained:

“Regarding cutting maneuver, subjects were instructed to run at medium-to-fast pace for 7 m before touching with their right foot the change of direction area and then to change the direction to the left (for left footed it was the opposite).”

Point 11: It may be helpful for the readers to follow the exact task that was performed while including some type of visual schematic with the layout of the task with actions performed.

Response 11: Cutting maneuver layout Figure 1 was added.

Point 12: What limb was used for the maximum angles?

Response 12: Details were added as follows:

“Regarding the landing maneuver, … Average of variables values of both lower limbs were used for the analysis. … Maximum knee and hip angle of dominant lower limb during this cutting maneuver at 45º were used for the analysis.”

Point 13: Line 205 - What is meant by low knee valgus angle? Recommend providing a reference for what the threshold would be for valgus angle. Revise sentence as grammar is confusing. Recommend removing "that too"

Response 13: Reference is #22.

“It was found that low knee valgus angle decreases risk of knee injuries [22].”

Point 14: One factor that is briefly mentioned but may be a main component is that the futsal players may not be used to the landing maneuver based on the fact they may not perform that action as often compared to the footballers. Although they may be at a greater risk for injury based on the measures you made, the action may not be relevant to the group in question and injury risk.

Response 14: Your point was taken into consideration as follows:

“Nevertheless, futsal players do not perform landing maneuver as often compared with football probably due to the different match-play (i.e., with more “low balls” over a smaller pitch). Therefore, landing maneuver may not be that relevant to futsal players’ injury risk given that this action is not as usual as in football.”

Point 15: Were any other performance measures taken such as jump height (mentioned but not reported), sprint speed, or lower body strength? There may be differences in these values and this could be the explanation for these differences observed between groups. There was a mention of 3 session per week of practice/competition but what about any type of resistance training or current injury prevention programs?

Response 15: Information was added.

Point 16: Based on the findings, was there any follow-up with the individuals that those were highlighted as being more likely to get injured and actually getting injured during sport practice/competition?

Response 16: We thank expert reviewer for his/her suggestion. We did not follow-up subjects for any injuries, but we have predicted injury based on modifiable intrinsic risk factors such as kinematics.

We hope that the manuscript has now reached the standard necessary for formal acceptance in Healthcare.

We look forward to hearing from you.

Best regards

Reviewer 2 Report

Comparison of lower extremities’ kinematics during landing and cutting between elite male football and futsal players

Introduction

L40 – what is classed as a high intensity phase (heart rate? distance covered? number of sprints?)

L44 – could specify the amount of lower limb injuries (%) for both football and futsal

L47 – is it worth noting the damage/ severity of ACL injuries in loss of training days or match games, rather than cost?

L55 – say lower limb injuries rather than just injuries

L71 – define what contributed to a safe landing is

L94 – replace ‘studied’ with ‘examined’ or ‘established’

Method

L109- state the level of players, similar to that in the abstract (L23)

L114- need to state how it determined when participants knew the ‘proper’ way to carry out the test. Did they need to meet a certain criterion? Or was it when participants were comfortable with the movements

L123- add measurements for heights and mass, i.e centimetres and kilograms

L138 – reliability measures for testing equipment      

L138 – add measurement for maximum jump height

L141- change the wording ‘like over’ with ‘similar to’

L142- how did you established a distance of 5m?

L143 – make sure writing is in the 3rd person, change ‘he’ to ‘the participants’

L150 – reliability of using 45 degrees

Discussion

L185 – state what made their landings safer

L188 – worth stating what why/ how these factors will impact landing manoeuvres in futsal

L189 – need to state why football requires more jumping and landing, i.e ball time in the air is greater in football then futsal

L196 – how does this relate to your findings? Need to show how this study relates to your current findings, does this support what you found or go against it.

Conclusion

L228- add lower limb injuries

L232 – mention exact training for injury prevention for futsal and football specifically  

Author Response

Does the introduction provide sufficient background and include all relevant references?

(x) Can be improved

Please, read below comments to specific points.

Is the research design appropriate?

(x) Can be improved

Please, read below comments to specific points.

Are the methods adequately described?

(x) Can be improved

Please, read below comments to specific points.

Are the results clearly presented?

(x) Can be improved

Please, read below comments to specific points.

Are the conclusions supported by the results?

(x) Can be improved

Please, read below comments to specific points.

Point 1: L40 – what is classed as a high intensity phase (heart rate? distance covered? number of sprints?).

L44 – could specify the amount of lower limb injuries (%) for both football and futsal.

Response 1: We thank expert reviewer for his/her suggestions. Futsal is a sport more intense than football in terms of more frequent non-contact injuries. This was stated as follows:

“Futsal is a multiple-sprint sport with more high intensity phases than football as witnessed by the higher relative number of injuries in futsal occurring during non-contact activities compared with football (37% vs. 27% [2]).”

Point 2: L47 – is it worth noting the damage/severity of ACL injuries in loss of training days or match games, rather than cost?

Response 2: Thanks for the wise suggestion that was operated adding following statement:

“Commonly, the players are allowed to return to sporting activities including cutting and jumping six months after ACL injuries. There is high risk of long-term osteoarthritis in all treated populations following ACL injury [6].”

Point 3: L55 – say lower limb injuries rather than just injuries.

Response 3: Suggestion was operated.

Point 4: L71 – define what contributed to a safe landing is.

Response 4: Details were added as follows:

“Modellistic estimate of “safe” (in terms of dynamics, i.e, subject’s maximum available braking torque ≤N·m; see Figure 9 from Minetti et al. [12]) landing was shown to happen when falling from a maximal height of 1.6-2.0 m in sedentary subjects.”

Point 5: L94 – replace ‘studied’ with ‘examined’ or ‘established’.

Response 5: Suggestion was operated.

Point 6: L109 - state the level of players, similar to that in the abstract (L23).

Response 6: Detail was added as follows:

“All subjects – with at least 3-year experience in Kerman Province League – were par-ticipating in football clubs with similar training schedules that included practice or competition at least 3 times per week (60-90 min per session).”

Point 7: L114 - need to state how it determined when participants knew the ‘proper’ way to carry out the test. Did they need to meet a certain criterion? Or was it when participants were comfortable with the movements?

Response 7: Detail was added as follows:

“On a separate occasion before the test session, all the players attended a workshop to learn the proper ways to carry out the tests, i.e., up to feeling comfortable with undergoing them.”

Point 8: L123- add measurements for heights and mass, i.e., centimetres and kilograms.

Response 8: Mass [kg] and height [m] values are already there in 2. Materials and Methods.

Point 9: L138 – reliability measures for testing equipment.

Response 9: Information was added.

Point 10: L138 – add measurement for maximum jump height.

Response 10: Information was added.

Point 11: L141- change the wording ‘like over’ with ‘similar to’.

Response 11: “over” was changed to “during”.

Point 12: L142- how did you established a distance of 5 m?

Response 12: Detail was added as follows:

“For this purpose, the subject was placed at a distance of 5 m from the ball (i.e., a reasonable run-up distance just before a heading following a corner).”

Point 13: L143 – make sure writing is in the 3rd person, change ‘he’ to ‘the participants’.

Response 13: Suggestion was operated.

Point 14: L150 – reliability of using 45 degrees.

Response 14: Detail was added as follows:

“Range between the angles of 35° and 55° with respect to the initial path were found and marked using a goniometer and two cones on the floor and subjects were instructed to perform the change of direction at the angle of 45° [19].”

Point 15: L185 – state what made their landings safer.

Response 15: To better explain the finding, the two relative sentences were joined as follows:

“It was shown that football players have greater hip and knee flexion, but less knee valgus angle during landing maneuver – and therefore safer landing – than futsal players.”

Point 16: L188 – worth stating what why/ how these factors will impact landing manoeuvres in futsal.

Response 16: Following sentence was added:

“All these peculiarities concur to increase non-contact injuries in futsal [2].”

Point 17: L189 – need to state why football requires more jumping and landing, i.e., ball time in the air is greater in football then futsal.

Response 17: Sentence was expanded as follows:

“Playing football requires more activities such as jumping vertically and heading the ball, because pitch is larger and therefore long passes – and consequently “high balls” – are more frequent.”

Point 18: L196 – how does this relate to your findings? Need to show how this study relates to your current findings, does this support what you found or go against it.

Response 18: Paragraph was removed.

Point 19: L228 - add lower limb injuries.

Response 19: Suggestion was operated.

Point 20: L232 – mention exact training for injury prevention for futsal and football specifically.

Response 20: We thank expert reviewer for his/her suggestion. Sentence was reworded as follows:

“The findings of this research can be helpful for coaches and trainers, who can design exact training for injury prevention for futsal and football specifically.”

We hope that the manuscript has now reached the standard necessary for formal acceptance in Healthcare.

We look forward to hearing from you.

Best regards

Reviewer 3 Report

The work entitled " Comparison of lower extremities’ kinematics during landing and cutting between elite male football and futsal players?" has an interesting approach for publication in Healthcare. But there are some questions of form that should be taken into account prior to consider this article for publication.

I enclose my suggestions for consideration by the authors.

  • Title:

The title reflects injuries in the lower extremities, but really the work focuses mainly on the evaluation of the knee and hip, this must be modified.

  • Abstract:

This section should reflect all parts of the work, but there is a lack of contextualization of the topic and description of the methodology.

  • Introduction:

In my opinion, this section must be reformulated, non-relevant topics are developed in breadth, ideas are repeated, it does not follow an adequate common thread. This makes for a long and confusing reading introduction.

For example, it would unify the paragraph that begins on line 44 with the paragraph that begins on line 56, and would place them in the position of the paragraph on line 56.

It is necessary to support the sentence on line 41-43 with a bibliography.

The manuscript does not state the clear objective of this work.

  • Methods:

This section is the section with the greatest weight of the work and it did not follow a clear, concise structure of the development of the work. also omits information that I consider important so that the results take greater value.

I recommend the authors the following scheme.

  1. Brief statement that collects the legal data of the work, bioethics committee, informed consent, Helsinki statement…
  2. Description of the sample, apart from the data reflected by the authors. I think they should include:
    1. Type of sports club, and its location.
    2. Club category and competitions
    3. How long the participants have been practicing the sport.
    4. Characteristics of the playing field.
  3. Placement protocol of the devices used for the measurement.
  4. Measurement description of the three parameters studied.
  5. Statistic analysis

  • Discussion

The authors should delve more deeply into the contents expressed in this part, it would greatly improve the manuscript and add value to the results.

I would also recommend addressing the following topics, and the possible relationship between their results;

  1. Could the type of field of competition be an influencing factor in the results?
  2. and the sporting gesture, is it the same?
  3. What training interventions would the authors guide to achieve improvements?

  • Conclusions

The conclusions are not strictly limited to answering the objective. This may be because the objective is not clearly defined.

Author Response

Does the introduction provide sufficient background and include all relevant references?

(x) Must be improved

Please, read below comments to specific points.

Is the research design appropriate?

(x) Can be improved

Please, read below comments to specific points.

Are the methods adequately described?

(x) Must be improved

Please, read below comments to specific points.

Are the results clearly presented?

(x) Can be improved

Please, read below comments to specific points.

Are the conclusions supported by the results?

(x) Can be improved

Please, read below comments to specific points.

Point 1: The title reflects injuries in the lower extremities, but really the work focuses mainly on the evaluation of the knee and hip, this must be modified.

Response 1: We thank expert reviewer for his/her suggestion. Title was modified as follows:

“Comparison of knee and hip kinematics during landing and cutting between elite male football and futsal players”

Point 2: Abstract:

This section should reflect all parts of the work, but there is a lack of contextualization of the topic and description of the methodology.

Response 2: Abstract was expanded in terms of contextualization of the topic and description of the methodology.

Point 3: It would unify the paragraph that begins on line 44 with the paragraph that begins on line 56, and would place them in the position of the paragraph on line 56.

Response 3: Suggestion was operated.

Point 4: It is necessary to support the sentence on line 41-43 with a bibliography.

Response 4: Suggestion was operated.

Point 5: The manuscript does not state the clear objective of this work.

The conclusions are not strictly limited to answering the objective. This may be because the objective is not clearly defined.

Response 5: Aim was made more explicit.

Point 6: I recommend the authors the following scheme.

Brief statement … Statistic analysis.

Response 6: Section was reworked following recommended scheme.

Point 7: I would also recommend addressing the following topics, and the possible relationship between their results:

  1. Could the type of field of competition … 3. … to achieve improvements?

Response 7: We thank expert reviewer for his/her suggestion. Discussion was expanded to address suggested topics.

We hope that the manuscript has now reached the standard necessary for formal acceptance in Healthcare.

We look forward to hearing from you.

Best regards

Round 2

Reviewer 1 Report

Lines 140-144 - Different color font

Line 149 - Spell out minutes

Line 156 - change "hanged" to "hung" or "placed"

Lines 165-166 - How would this rather subjective interpretation of effort effect intensity, therefore impacting reliability and validity? Is there previous research that has used this method while reporting the reliability?

Author Response

Does the introduction provide sufficient background and include all relevant references?

(x) Can be improved

Please, read below comments to specific points.

Is the research design appropriate?

(x) Can be improved

Please, read below comments to specific points.

Are the methods adequately described?

(x) Can be improved

Please, read below comments to specific points.

Are the results clearly presented?

(x) Can be improved

Please, read below comments to specific points.

Are the conclusions supported by the results?

(x) Can be improved

Please, read below comments to specific points.

Point 1: Lines 140-144 - Different colour font.

Response 1: We thank expert reviewer for his/her remark. Mistake was amended.

Point 2: Line 149 - Spell out minutes.

Response 2: Correction was operated.

Point 3: Line 156 - change "hanged" to "hung" or "placed"

Response 3: Correction was operated.

Point 4: Lines 165-166 - How would this rather subjective interpretation of effort effect intensity, therefore impacting reliability and validity? Is there previous research that has used this method while reporting the reliability?

Response 4: We thank expert reviewer for his/her suggestion. Sentence was re-worked.

We hope that the manuscript has now reached the standard necessary for formal acceptance in Healthcare.

We look forward to hearing from you.

Best regards

Reviewer 3 Report

Thank the authors for the work done to improve the manuscript entitled " Comparison of lower extremities’ kinematics during landing and cutting between elite male football and futsal players?" obtaining a suitable manuscript and with an interesting approach for its publication in Healthcare.

Author Response

Point 1: Thank the authors for the work done to improve the manuscript entitled " Comparison of lower extremities’ kinematics during landing and cutting between elite male football and futsal players?" obtaining a suitable manuscript and with an interesting approach for its publication in Healthcare.

Response 1: We thank expert reviewer for his/her appreciation.

We hope that the manuscript has now reached the standard necessary for formal acceptance in Healthcare.

We look forward to hearing from you.

Best regards